# Identification of Potential Biomarkers for Anti-PD-1 Therapy in Melanoma by Weighted Correlation Network Analysis

**DOI:** 10.3390/genes11040435

**Published:** 2020-04-17

**Authors:** Xuanyi Wang, Zixuan Chai, Yinghong Li, Fei Long, Youjin Hao, Guizhi Pan, Mingwei Liu, Bo Li

**Affiliations:** 1Key Laboratory of Clinical Laboratory Diagnostics, College of Laboratory Medicine, Chongqing Medical University, Chongqing 400046, China; xuanyi_w@163.com (X.W.); bioczx@stu.cqmu.edu.cn (Z.C.); leonardfei@163.com (F.L.); panpan.helen@gmail.com (G.P.); 2School of Biological Information, Chongqing University of Posts and Telecommunications, Chongqing 400065, China; liyinghong@cqupt.edu.cn; 3College of Life Sciences, Chongqing Normal University, Chongqing 401331, China; haoyoujin@hotmail.com

**Keywords:** melanoma, anti-PD-1 therapy, WGCNA, biomarker

## Abstract

Melanoma is the most malignant form of skin cancer, which seriously threatens human life and health. Anti-PD-1 immunotherapy has shown clinical benefits in improving patients’ overall survival, but some melanoma patients failed to respond. Effective therapeutic biomarkers are vital to evaluate and optimize benefits from anti-PD-1 treatment. Although the establishment of immunotherapy biomarkers is well underway, studies that identify predictors by gene network-based approaches are lacking. Here, we retrieved the existing datasets (GSE91061, GSE78220 and GSE93157, 79 samples in total) on anti-PD-1 therapy to explore potential therapeutic biomarkers in melanoma using weighted correlation network analysis (WGCNA), function validation and clinical corroboration. As a result, 13 hub genes as critical nodes were traced from the key module associated with clinical features. After receiver operating characteristic (ROC) curve validation by an independent dataset (GSE78220), six hub genes with diagnostic significance were further recovered. Moreover, these six genes were revealed to be closely associated not only with the immune system regulation, immune infiltration, and validated immunotherapy biomarkers, but also with excellent prognostic value and significant expression level in melanoma. The random forest prediction model constructed using these six genes presented a great diagnostic ability for anti-PD-1 immunotherapy response. Taken together, *IRF1*, *JAK2*, *CD8A*, *IRF8*, *STAT5B,* and *SELL* may serve as predictive therapeutic biomarkers for melanoma and could facilitate future anti-PD-1 therapy.

## 1. Introduction

Melanoma, as a form of highly aggressive skin cancer, is easy to metastasize and thus difficult to treat [1]. Recent clinical studies with anti-PD-1 immunotherapy have shown superior clinical efficacy and significant survival benefits for melanoma patients [2]. However, only a portion of the patients have an objective response to PD-1 blocking immunotherapy [3], and the remaining ones show little or no response, even involving in a high-grade immune-related adverse event [2,4,5]. Moreover, the use of immunotherapy would greatly increase the medical cost for an individual patient [6,7]. Regarding the clinical response, costs and side effects, it is urgent that predictive biomarkers are needed for assistance in identifying which patients are more prone to benefit from checkpoint inhibitor-based immunotherapy.

To date, it has been noticed that the clinical efficacy of anti-PD-1 therapy is relevant to the intrinsic features of tumor cells and tumor microenvironment (TME) or gene signatures. Some characteristics such as tumor mutational burden, deficiency of mismatch repair, host expression of PD-1/PD-L1 and the density of tumor-infiltrating lymphocytes have been reported as promising biomarkers for anti-PD-1 therapy [8]. Of which, the expression of PD-L1 typifies the most well-focused potential biomarker for the anti-PD-1 immune checkpoint blockade therapy response [9]. Several studies have revealed that PD-L1 expression played a crucial role in enriching the anti-tumor response for pre-treatment melanoma specimens, and 30–40% of melanoma patients with high expression levels of PD-L1 had durable and objective responses [10]. The clinical research by Cottrell, T.R. et al. addressed a similar result [11]. 

However, PD-L1 expression is dynamic and varies over time, which leads to poor anti-PD-1 efficacy [8]. For instance, Patel, S.P. et al. proposed that during melanoma, lung, kidney, and other cancers, over 36% of patients with positive PD-L1 IHC expression responded to the PD-1/L1 axis-directed therapy, but approximately 17% of patients with a negative expression were also responded [12]. Johnson, D.B. et al. found that PD-L1 expression had no predictive power for survival, but the interaction of PD-1/PD-L1 and IDO-1/HLA-DR co-expression may improve outcomes of anti-PD-1 therapy in melanoma [13].

Previous studies indicated that exploring tumor type-specific gene expression, dynamic omics profiles in intratumoral heterogeneity, the tumor cell infiltrate, and the tumor–host microenvironment was probably helpful for the establishment of anti-PD-1 therapeutic biomarkers [14,15,16]. For example, with genomic arrays from 14,492 distinct solid tumors, Messina, J.L. et al. discovered a novel expression signature of 12 chemokine genes that lead to a potentially suitable selection for improving immunotherapy in melanoma [17]. Ayers, M. et al. quantified T cell-inflamed gene-expression profiles in the microenvironment, and it was developed for pembrolizumab trials in melanoma [18]. Ribas, A. et al. confirmed that the expression signatures of interferon-inflammatory immune genes were related to the overall response rate in patients treated with pembrolizumab [19]. By evaluating the gene signatures of six immune cells, Varn, F.S. et al. found that the B cell-derived expression signature could predict the checkpoint inhibitor-based immunotherapy response for patients [20]. 

Although several studies have a start on biomarkers for anti-PD-1 benefits in melanoma, few of them focused on the functional correlation between the genes as well as the relationship between gene expressions and therapeutic response. The above studies limited the development and exploration of biomarkers and the molecular mechanisms to a systems biology perspective [21,22]. Therefore, a robust analysis based on omics data and gene co-expression network was applied in this study to provide an insight into the correlation among genes as well as between the genes and therapeutic response, which lead to the identification of potential therapeutic biomarkers of anti-PD-1 immunotherapy in melanoma.

In this study, all the existing RNA-seq datasets on anti-PD-1 immunotherapy of melanoma were reanalyzed using a bioinformatics approach, including GSE91061, GSE78220 and GSE93157 (79 samples in total). With the weighted correlation network analysis, the dataset of GSE91061 was adopted to construct gene co-expression network across the different samples, followed by the identification of key modules with therapeutic response. Functional enrichment analysis was further used to investigate the biological function of the key module genes. From this, the hub genes were identified based on the co-expression network, protein–protein interaction (PPI) network and gene topological network. GSE78220, as an independent dataset, was utilized to investigate the potential diagnostic genes for anti-PD-1 therapy from hub genes, which were plotted by receiver operating characteristic (ROC) analysis. Afterward, function enrichment, immune infiltration level, gene expression level, overall survival, and gene correlation of potential diagnostic genes were validated by Gene Set Variation Analysis (GSVA), Tumor Immune Estimation Resource (TIMER), and Gene Expression Profiling Interactive Analysis (GEPIA). A prediction model of immunotherapy was constructed by the hub genes based on random forest classifier. The accuracy of the prediction model was verified based on GSE93157.

## 2. Materials and Methods 

The workflow used in the study is shown in Figure 1, which included three main steps, hub gene identification, diagnostic efficiency analysis and validation of potential therapeutic biomarkers.

### 2.1. Data Acquisition of Gene Expression Datasets

Using the keywords such as “PD-1”, “immunotherapy”, “therapy”, “treatment”, and “melanoma”, datasets were investigated from the main transcriptomics database, including Gene Expression Omnibus (GEO) (http://www.ncbi.nlm.nih.gov/geo) [23], ArrayExpress (https://www.ebi.ac.uk/arrayexpress) [24] and Expression Atlas (https://www.ebi.ac.uk/gxa/home) [25]. By manually checking, the raw datasets meeting the following standards were retained in subsequent analyses: (i) inclusion of gene expression data of responders (complete response or particle response) or non-responders (progress disease) to anti-PD-1 therapy in melanoma. The responders and non-responders to immunotherapy were defined according to iRECIST guidelines [26], and the stable disease was not included due to the controversial role in response to therapy [27,28]; (ii) at least 15 accessible samples in the datasets; and (iii) availability of raw sequence or microarray data. 

### 2.2. Construction of Gene Co-Expression Network

The Dataset GSE91061 [27] (10 complete response or particle response samples and 23 non-response samples) was normalized with DESeq2 [29], and the genes were ranked by median absolute deviation (MAD). The top 5000 genes with the highest MADs were extracted for gene co-expression network construction via the weighted correlation network analysis (WGCNA) package [30]. Pearson’s correlation coefficients were calculated between each pair of the extracted genes to generate the adjacency matrix. Then, the function “tomlikeity” was used to transform the adjacency matrix into the topological overlap measure (TOM). The TOM reflected the correlative interconnectivity between two genes according to their degree of shared adjacency for the whole network [30]. The genes with similar expression patterns were clustered into the same module utilizing the average linkage hierarchical clustering based on the TOM-based dissimilarity measure.

### 2.3. Identification of Clinically Significant Modules

Two approaches were utilized to obtain the modules significantly associated with clinical traits. Pearson’s correlation analysis was adopted to calculate the correlation between clinical features and the module eigengenes (MEs). MEs were the major component for each gene module and the most representative expression patterns of the module [30]. Then, the gene significance (GS) and module significance (MS) were calculated. GS represented the correlation between gene expression and clinical features. MS was the average GS across all genes in the module [30]. Generally, the module with the first-ranked MS was considered as the clinically significant modules.

### 2.4. Gene Ontology (GO) and Kyoto Encyclopedia of Genes and Genomes (KEGG) Pathway Enrichment Analysis

To investigate the function of the genes in clinically significant modules, enrichment analysis of Gene Ontology (GO) was performed by NetworkAnalyst (https://www.networkanalyst.ca) [31]. GO terms were regarded as the significant ones when the adjusted *p*-value < 5 × 10^−3^ and minimum gene counts > 20. NetworkAnalyst was also applied for performing Kyoto Encyclopedia of Genes and Genomes (KEGG) pathway enrichment analysis. Adjusted *p*-value < 2 × 10^−4^ and the minimum gene counts > 10 were regarded as the cut-off criteria. 

### 2.5. Identification of Hub Genes Based on Protein–Protein Interaction (PPI) and Topological Network

The PPI network of key module genes was constructed by the STRING database (http://string-db.org) [32]. The plug-in CytoHubba [33] of Cytoscape software [34] was applied to explore key nodes in the PPI network via 11 topological algorithms including Degree, Edge Percolated Component, Maximum Neighborhood Component, Density of Maximum Neighborhood Component, Maximal Clique Centrality, Bottleneck, EcCentricity, Closeness, Radiality, Stress, and Betweenness [33]. The top 50 nodes were defined as core genes for each algorithm in the topological network. The intersected core genes derived from the 11 topological algorithms were considered as the hub genes with important biological regulatory functions.

### 2.6. Receiver Operating Characteristic (ROC) Curve Analysis 

To verify the diagnostic ability of hub genes in another dataset, the RNA-sequencing dataset GSE78220 [28] was utilized, which included 28 melanoma patient samples (five complete response samples, 10 partial response samples and 13 non-response samples) treated with anti-PD-1 therapy. Using pROC package [35], the area under the ROC curve (AUC) was calculated on the expression data of each hub gene. In addition, to obtain test results with high specificity, we focused on the partial area under the curve (pAUC) between 90% and 100% specificity [36], which was also calculated by the pROC package [35]. In the present study, the larger AUC or pAUC value for a gene indicated that this gene can better distinguish responders from non-responders for anti-PD-1 immunotherapy [37]. Based on the AUC or pAUC values, the diagnosis effect of hub genes was further investigated.

### 2.7. Gene Set Variation Analysis of Hub Genes

GSVA is a reliable approach to evaluate the variation of function activity across different samples via an unsupervised manner [38]. To explore the functions most associated with hub genes, “GSVA” R package was applied based on the GSE78220 [28]. The gene set “c5.all.v2.5.symbols.gmt” was downloaded from the Molecular Signature Database (MSigDB) accessed on 5 December 2019 (http://software.broadinstitute.org/gsea/msigdb/index.jsp) as the reference gene set, and *p*-value < 0.05 was used as the cut-off point.

### 2.8. Correlation Analysis of Hub Genes and Immune Infiltration Level 

The online tool TIMER (https://cistrome.shinyapps.io/timer) database [39] was used to explore the association between the hub gene expression and immune infiltration levels in melanoma. Based on deconvolution of the previously published statistical methods [40], the TIMER database is a comprehensive resource to evaluate the abundance of tumor-infiltrating immune cells (TIICs) across diverse cancer types from The Cancer Genome Atlas (TCGA) database (https://cancergenome.nih.gov).

### 2.9. Validation and Survival Analysis of Hub Genes

GEPIA database (http://gepia.cancer-pku.cn/index.html) provides comprehensive expression analyses for RNA sequencing data, including 9,736 tumors and 8,587 normal samples from TCGA and the GTEx projects [41]. To reveal the expression level and prognostic value of hub genes in melanoma, differential expression analysis (*p*-value < 0.05 and log_2_FC > 1) and survival analysis (*p*-value < 0.05) were performed via GEPIA. In addition, the Cox proportional hazard regression model of the screened hub genes was constructed via the survival package in R software to evaluate the overall survival of the melanoma patients. The Cox proportional hazard regression model is a very useful tool to access the impact of lifetime-related factors on the hazard function [42]. The skin cutaneous melanoma (SKCM) and uveal tract melanoma (UVM) datasets from TCGA as well as normal samples from GTEx projects were used for analysis.

### 2.10. Correlation Analysis of Hub Genes and Biomarkers of Anti-PD-1 Therapy

GEPIA was also utilized to identify the relationship of hub genes and biomarkers of anti-PD-1 immunotherapy. The SKCM and UVM datasets from TCGA were used for analysis. The Spearman method was adopted to calculate the correlation coefficient between genes. The terms with *p*-value < 0.05 were regarded as statistically significant.

### 2.11. Random Forest

A prediction model of anti-PD-1 immunotherapy response was constructed via the random forest classifier. The hub genes were the covariates of the prediction model. The random forest is a popular tool for classification and regression, which shows a powerful ability to construct a predictive model for new biomarkers. The random forest is less prone to over-fitting problems and can handle a large amount of noise. A random forest-based classifier was built via the randomForest package in R software based on the algorithm of Breiman and Cutler [43]. The samples of GSE78220 (*n* = 28) [28] were randomly divided into the training set and test set via the caret package, each of which contained 14 samples. Then, the decision tree model of the training set was established to obtain the classification. Next, the classification results of each time were averaged to calculate the final classification. The model built by the training set would be tested by the test set. Each result would calculate the error rate through Out-of-bag (OOB) to evaluate the correct rate of the combined classification. OOB was the data not sampled when the training set was randomly sampled. The OOB samples were used to estimate the prediction error and variable importance [44]. Finally, the melanoma samples treated with anti-PD-1 therapy of GSE93157 [45] (seven complete response or particle response samples and 11 non-response samples) were used as the validation set to verify the accuracy of the random forest model. AUC index was utilized to evaluate the efficiency of the prediction model.

## 3. Result

### 3.1. Construction of Weighted Co-Expression Network and Identification of Key Modules

According to the strict standards described above, GSE91061 (*n* = 33) [27], GSE78220 (*n* = 28) [28] and GSE93157 (*n* = 18) [45] were retained for further analysis. To determine the key modules connected with clinical features (therapeutic response), the weighted co-expression network was constructed by WGCNA based on the GSE91061 (*n* = 33). The power of β = 5 (scale free *R*^2^ = 0.89) was selected as the soft-thresholding parameter to ensure a scale-free network (Figure 2A). A total of 27 modules, ranging in size from 30 to 525 genes, was found by the average linkage hierarchical clustering (Figure 2B–C). Based on the calculation of Pearson’s correlation coefficient, the pink module was considered as the highest correlation one with the clinical traits (cor = 0.38, *p*-value = 0.03) (Figure 2D). Moreover, the pink module showed the highest MS in Figure 2E. Modules with a greater MS were relevant to the clinical traits. According to the results of the two methods, the pink module was selected as a key module to be studied in subsequent analyses.

### 3.2. Potential Functional Roles of Genes in the Pink Module

The results showed 19 GO terms were enriched with adjusted *p*-value < 5 × 10^-3^ and minimum gene counts > 20 (Figure 3A). The pink module genes were mainly enriched in immune function (e.g., immune response, immune progress, and immune cell activation), hemopoiesis and signal transduction. Activating the immune system via blocking the immune checkpoint was a crucial factor for anti-PD-1 immunotherapy to attack the tumor cells [46], which indicated that genes involved the immune function may be relevant to the efficacy of anti-PD-1 immunotherapy.

Based on KEGG pathway mapping, the pink module genes were significantly involved in 12 pathways (adjusted *p*-value < 2 × 10^-4^, minimum gene counts > 10) such as Epstein–Barr virus infection, chemokine signaling pathway, natural killer cell-mediated cytotoxicity, Th1 and Th2 cell differentiation, Jak-STAT signaling pathway, and so on (Figure 3B). Natural killer cell-mediated cytotoxicity was the fourth most significant pathway (adjusted *p*-value = 4.33 × 10^-5^). Ardolino, M. et al. found that the natural killer cell response elicited by the PD-1/PD-L1 blockade played vital roles in the therapeutic effect of immunotherapy [47]. The results also demonstrated that many genes were significantly involved in the chemokine signaling pathway. Chemokine gene expression signatures including *CCL5*, *CXCL9*, *CXCL10,* and *CXCL11* were reported that could accurately predict anti-PD-1 immunotherapy response for patients with head and neck squamous cell carcinoma and gastric cancer [18]. Furthermore, Herbst, R.S. et al. found that the expression of *CXCL9* had a significant, positive correlation with the therapeutic response in melanoma [48]. Comparing melanoma samples with normal controls, Boots, A.M. et al. indicated that PD-1 checkpoint blockades enhanced the inflammatory responses of Th1 and Th17 as well as inhibited Th2 responses [49]. Genes involved in Th1 and Th2 cell differentiation may indirectly reflect the response of blockades. Additionally, the Jak-STAT signaling pathway was also enriched. Lu, C. et al. demonstrated that Jak-STAT signaling inhibited cytotoxic T lymphocyte activation to weaken the effect of anti-PD-1 immunotherapy [50]. 

### 3.3. Identification of Hub Genes 

A total of 232 genes in the pink module were analyzed by STRING database. A PPI network containing 100 nodes and 134 interactions was built with the medium confidence score (> 0.4). After importing the data into Cytoscape and running the CytoHubba program, the top 50 node genes were calculated as the core genes by 11 topological algorithms, respectively (Appendix A). As a result, a total of 13 genes including *BTK*, *CD3E*, *CD48*, *IL2RG*, *IL2RB*, *LCP1*, *TRIM21*, *IRF1*, *JAK2*, *CD8A*, *IRF8*, *STAT5B,* and *SELL* were calculated as the intersection of the core genes in 11 algorithms, which were considered to be hub genes.

### 3.4. ROC Curve Analysis of Hub Genes

To validate the predictive power of 13 hub genes for anti-PD-1 therapy in melanoma, ROC curve analysis was enabled utilizing GSE78220. Finally, the results suggested that the expression of six genes including *IRF1*, *JAK2*, *CD8A*, *IRF8*, *STAT5B,* and *SELL* had a significant ability to distinguish the responders from non-responders to anti-PD-1 therapy in melanoma with AUC > 0.6 and pAUC > 0.7 (Figure 4). 

### 3.5. Gene Expression of Hub Genes 

The expression of the six genes in responders or non-responders to anti-PD-1 therapy (GSE91061) was shown in Figure 5. The expressions of *IRF1*, *JAK2*, *CD8A*, *IRF8,* and *SELL* were down-regulated in the responders compared with non-responders (*p*-value < 0.05 and log_2_FC > 0.5). However, the expression of *STAT5B* had no significant changes between the responders and non-responders.

### 3.6. Functional Enrichment Analysis by Gene Set Variation Analysis (GSVA) 

To further investigate the correlation between anti-PD-1 immune checkpoint blockade therapy and functional features of *IRF1*, *JAK2*, *CD8A*, *IRF8*, *STAT5B,* and *SELL* in melanoma, differential signature enrichment analysis was performed via GSVA based on the GSE78220 datasets (*n* = 28). A group of 27 functional signatures was enriched in RNA-seq data of 13 non-responding versus 15 responding pre-anti-PD-1 melanoma patients. The enrichment functions included regulation of transcription factor and promoter, lymphocyte proliferation and differentiation, regulation of hematopoiesis and the immune system, and so on (Figure 6). Most of the functions were up-regulated in the group with positive responses to immunotherapy and down-regulated in the group with negative responses to immunotherapy. This demonstrated that the functional signatures of *IRF1*, *JAK2*, *CD8A*, *IRF8*, *STAT5B,* and *SELL* could provide several suggestions for distinguishing the clinical effects of anti-PD-1 immunotherapy.

### 3.7. Analysis of Association Between Hub Genes and Immune Infiltration Level

Recent studies confirmed that immune infiltrating lymphocytes, as a crucial factor in regulating the immune system, had vast potential to predict the effect of checkpoint inhibitor therapy [7]. In this study, the TIMER database was applied to analyze the relationship between the hub gene expressions and immune infiltration levels in melanoma. As a whole, the expressions of six hub genes in SKCM were associated with B cells, CD4^+^ T cells, CD8^+^ T cells, neutrophils, macrophages, and dendritic cells (*p*-value < 0.05) (Figure 7A–F). The overall trend of these six genes was negatively related to tumor purity, which demonstrated that these genes may be highly expressed in TME. The expressions of *IRF1*, *CD8A* and *IRF8*, especially *IRF8*, were significantly correlated with the infiltration of CD8^+^ T cells, neutrophils and dendritic cells (cor > 0.5, *p*-value < 1 × 10^-5^). Several studies revealed that CD8^+^ T cells, neutrophils and dendritic cells were related to better outcomes and longer survival for patients under immunotherapy [51,52,53], which implied *IRF1*, *CD8A* and *IRF8* were probably connected with the prognostics of immune treatment. However, the correlation between the hub genes expression and immune infiltrating levels was not obvious in UVM.

### 3.8. Validation of Hub Genes in The Cancer Genome Atlas (TCGA) Datasets

The prognostic value and expression level of the six hub genes were validated by TCGA datasets on melanoma. The results indicated that *CD8A* and *SELL* were up-regulated as well as *JAK2* was down-regulated in melanoma samples compared with normal controls (*p*-value < 0.05) (Figure 8A). Furthermore, the results of the survival analysis suggested that high expressions of *IRF1*, *JAK2*, *CD8A*, *IRF8,* and *SELL* correlated significantly with improved clinical outcomes (*p*-value < 0.01) (Figure 8B). In addition, the multivariate Cox regression analyses for the six genes including *IRF1*, *JAK2*, *CD8A*, *IRF8*, *STAT5B,* and *SELL* were performed. The Cox proportional hazard regression model equation is as follows:*Risk_a_* = −0.127546 · *IRF1* − 0.138979 · *JAK2* −; 0.098595 · *CD8A* + 0.132417 · *IRF8* − 0.070296 · *STAT5B* − 0.009357 · *SELL*(1)

The risk score was calculated for each sample, and the samples were grouped according to the median risk score (cutoff = –3.376457). The results showed that the prognoses of the high-risk and low-risk groups significantly differed (Figure 8C). High expression of *IRF8* was related to a high risk of death. The high expression of *IRF1*, *JAK2*, *CD8A*, *STAT5B,* and *SELL* was connected with a low risk of death and was a protective factor of melanoma. The results were roughly consistent with the survival analysis of a single gene. In previous studies, anti-PD-1 immunotherapy usually showed a remarkable ability to effectively improve the survival benefits of patients across several cancers [54]. Patients with favorable prognosis frequently showed a positive response to anti-PD-1 therapy [55]. Hence, combining with the results of survival and Cox regression analyses, it implied that the high expressions of *IRF1*, *JAK2*, *CD8A,* and *SELL* may predict a positive response to anti-PD-1 immunotherapy for melanoma patients.

### 3.9. Relationships between Hub Genes and Biomarkers of Anti-PD-1 Therapy

The relevance of gene expression between the six hub genes and the reported biomarkers (PD-L1/*CD274*, *CXCR3* and IFN-γ/*IFNG*) with significant predictive power for immunotherapy were analyzed by the GEPIA database. The expression level of PD-L1 IHC in tumor cells has been indicated as a unique biomarker of the immune checkpoint blockade response in the clinical [56]. *CXCR3* and IFN-γ have been confirmed by more than one article as biomarkers for sensitivity to the PD-1 blockade based on clinical experiments and mouse models [18,57,58,59]. The results of correlation analysis showed a significant, positive correlation (cor > 0.6, *p*-value < 0.05) between the expression of four hub genes (*IRF1*, *CD8A*, *IRF8,* and *SELL*) and three biomarkers in melanoma (Figure 9A–C). Genes with strong correlation may have similar regulatory capacities or biological functions [60]. Thus, we speculated that the four genes closely related to PD-L1, IFN-γ and *CXCR3* may be connected with the efficacy evaluation of anti-PD-1 immunotherapy. 

### 3.10. The Random Forest Model of Hub Genes

We constructed a random forest classification model of anti-PD-1 immunotherapy response based on the screened six hub genes (Appendix A). During the process of building the random forest model, when mtry = 3, the false positive rate of the model was the lowest (Figure 10A). The optimal model can be achieved when the number of decision trees was about 2,000 (Figure 10B). In addition, the randomForest package provides two indexes to calculate the importance of variables. The one is the index to calculate the prediction error rate based on OOB and is named mean decrease accuracy (%IncMSE). The other is to calculate the Gini coefficient based on the sample fitting model and is named Mean Decrease Gini (IncNodePurity). The results showed that *IRF1* and *JAK2* were the more important variables in the prediction model (Figure 10C). Then, the AUC index was used to evaluate the efficiency of the model. The results showed that the prediction model had a good predictive ability for anti-PD-1 immunotherapy response (AUC = 0.75) (Figure 10D). Compared with the single gene, the random forest model had a better value of AUC except for *IRF1*. Additionally, the samples of an independent dataset GSE93157 (*n* = 18) were used as the validation set to verify the accuracy of the random model. The results also indicated that the random forest model could significantly distinguish the response to anti-PD-1 therapy for melanoma patients (AUC = 0.71) (Figure 10E).

## 4. Discussion

Despite the significant survival benefits of applying PD-1 checkpoint blockades in patients with melanoma, the response is only observed in a small group, and a massive economic burden comes to the individual patient [51]. Currently, the PD-L1 expression upon IHC is the unique biomarker approved in clinical practice, while its role is still controversial [61]. Therefore, the establishment of predictive biomarkers for immunotherapy response has become a priority.

To investigate potential biomarkers for anti-PD-1 immune checkpoint blockade therapy in melanoma, omics data and network-based approaches were applied in the study. Utilizing gene expression data, a gene co-expression network was built via the WGCNA algorithm to identify key modules related to the clinical features (therapeutic response) [30]. The core genes of the clinically significant modules were supposed to be crucial genes in the occurrence and development of disease [62]. Eventually, the pink module was screened, and 13 hub genes were derived from the module. Furthermore, ROC curve analysis showed that six of 13 hub genes including *IRF1*, *JAK2*, *CD8A*, *IRF8*, *STAT5B,* and *SELL* can specifically and accurately distinguish responders from non-responders to anti-PD-1 therapy, which implied the six genes appeared as potential predictors.

To further explore the biological functions of the six hub genes, functional enrichment analysis, survival analysis, immune infiltration level analysis, correlation analysis, and prediction model analysis were enabled. Our results demonstrated that the six hub genes were enriched in lymphocyte proliferation and differentiation as well as immune system regulation. These enriched biological functions contributed to the positive regulation of the immune system and the promotion of anti-tumor immune response [63]. Moreover, the results of survival analysis indicated that the high expression of six hub genes except *STAT5B* was connected with improved outcomes in melanoma. The multivariate Cox regression analysis implied that high expression of *IRF8* was associated with a high risk of death, and high expressions of *IRF1*, *JAK2*, *CD8A*, *STAT5B,* and *SELL* were associated with low risk of death. Patients with excellent prognosis frequently revealed positive reactiveness to anti-PD-1 therapy [54]. Thus, we speculated that the high expressions of *IRF1*, *JAK2*, *CD8A,* and *SELL* might be related to the favorable response of immunotherapy. Besides, our results showed that the expression level of the six hub genes was relevant to TIICs. As a valued factor of anti-tumor immunity, the TIICs can impact the prognosis of patients under immunotherapy [56]. Finally, four genes (*IRF1*, *CD8A*, *IRF8,* and *SELL*) were found that closely related to biomarkers of anti-PD-1 therapy (PD-L1, IFN-γ and *CXCR3*). The strong correlation between genes implied that genes may have similar biological functions or regulatory effects [60]. This suggested that the four genes may have a similar ability to predict the anti-PD-1 immunotherapy response. Besides, the random forest prediction model constructed based on the six hub genes showed significant prediction ability of anti-PD-1 therapy. Together with the above findings, there is a strong support of these six hub genes playing pivotal roles in the prognosis and response of checkpoint blockade immunotherapy. 

Notably, previous studies provided a large amount of evidence to support the reliability of the results in this study. Rimm, D.L. et al. found that *IRF-1* expression was higher in melanoma patients with partial or complete response to anti-PD-1 therapy based on the clinical experiments [64]. Shin, D.S. et al. proposed that *JAK2* loss-of-function mutations lead to resistance to PD-1 blockade therapy due to the lack of reactive PD-L1 expression and response to interferon-gamma [65]. Moreover, *IRF1* and *JAK2* had been reported as biomarkers of the anti-PD-1 immunotherapy response in melanoma via clinical sample validation [64,66]. The *CD8A* is a cell surface glycoprotein on most cytotoxic T lymphocytes, which can mediate efficient cell–cell interactions within the immune system [67]. Wherry, E.J. et al. demonstrated an association between *CD8A* expression in tumors and response to immune checkpoint inhibitors in melanoma [68]. Meanwhile, *CD8A* was conformed as a biomarker to predict the clinical effects of nivolumab in lung cancer [69]. The remaining three hub genes were considered to be closely related to immunotherapy or the immune response. The expression of *IRF8* can selectively induce and maintain the production of soluble factors to regulate the immune response [70]. *IRF8* contributed to antitumor immunity due to promoting the differentiation of CD4^+^ cells and CD8^+^ cells as well as activation of natural killer cells [71]. *STAT5* belongs to the T-cell transcription factor family, which contains two highly related proteins, *STAT5A* and *STAT5B* [72]. Auphan-Anezin, N. et al. found targeting *STAT5* in tumor-associated immune cells increased the clinical benefits of immunotherapy [73]. Finotto, S. et al. demonstrated that the function of *STAT5* can help to improve the effect of lung cancer immunotherapy via detecting the gene expression of *STAT5* in CD4^+^ T lymphocytes [72]. Majri, S.S. et al. described *STAT5B* was a crucial regulator of restimulation-induced T cell death in humans and mice [74]. *SELL* is a cell adhesion molecule on leukocytes and the preimplantation embryo, which plays crucial roles in lymphocyte–endothelial cell interactions [75]. It was confirmed that *SELL* can improve the efficacy of cancer immunotherapy via enhancing the activity of T cells [76].

Although the expression of *STAT5B* showed no significant changes between the responders and non-responders to anti-PD-1 therapy based on GSE91061, the gene should not be ignored. *STAT5B* was the hub gene identified by WGCNA, which may play an important role in the gene network of immunotherapy. ROC analysis indicated that it had significant diagnostic value in the anti-PD-1 therapy response (AUC = 0.72, pAUC = 0.80). *STAT5B* can combine with the other five screened hub genes to construct a random forest prediction model of immunotherapy response with excellent diagnostic efficiency. The results of functional enrichment analysis, immune infection level analysis, correlation analysis and literature validation demonstrated *STAT5B* was closely associated with the evaluation of response to anti-PD-1 therapy. Therefore, *STAT5B* was worthy to regard as a potential therapeutic biomarker.

Based on the random forest tool, a prediction model of six genes was constructed to detect the diagnostic efficacy of anti-PD-1 immunotherapy. The results showed that the random forest model of the six genes can distinguish patients that have a response to immunotherapy. However, it is worth noting that not too many samples were used to train the prediction model at present. If the model was directly applied to other datasets, the prediction efficiency may not be satisfactory enough. Therefore, in the future, we will collect more datasets associated with anti-PD-1 therapy in melanoma and continue to retrain our prediction model so that it can be directly extended to other studies. In addition, other limitations of the study were that the six hub genes identified by WGCNA lacked experimental verification. In particular, the results in the present study need to be verified by clinical trials. 

In conclusion, six hub genes (*IRF1*, *JAK2*, *CD8A*, *IRF8*, *STAT5B,* and *SELL*) were discovered to distinguish the response of melanoma patients under anti-PD-1 immunotherapy via WGCNA and integrated bioinformatics. These genes may serve as potential biomarkers to guide immunotherapy in the future.

## Figures and Tables

**Figure 1 genes-11-00435-f001:**
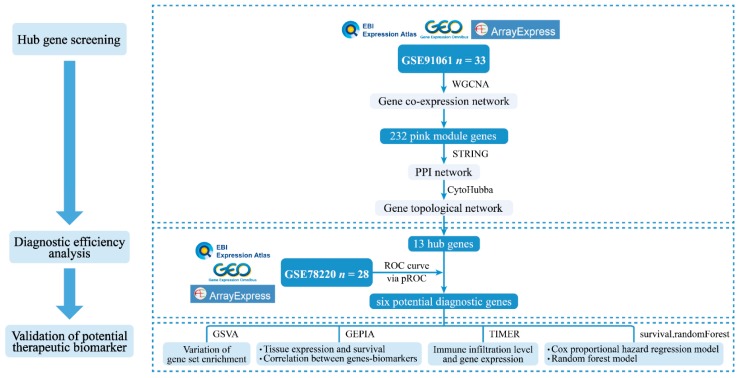
The flowchart of study design. Flow diagram of analysis procedure including discovery, analysis and validation of potential therapeutic biomarkers in this study. GEO: Gene Expression Omnibus; WGCNA: Weighted correlation network analysis; PPI: Protein–protein interaction; ROC: Receiver operating characteristic; GSVA: Gene set variation analysis; GEPIA: Gene expression profiling interactive analysis; TIMER: Tumor immune estimation resource.

**Figure 2 genes-11-00435-f002:**
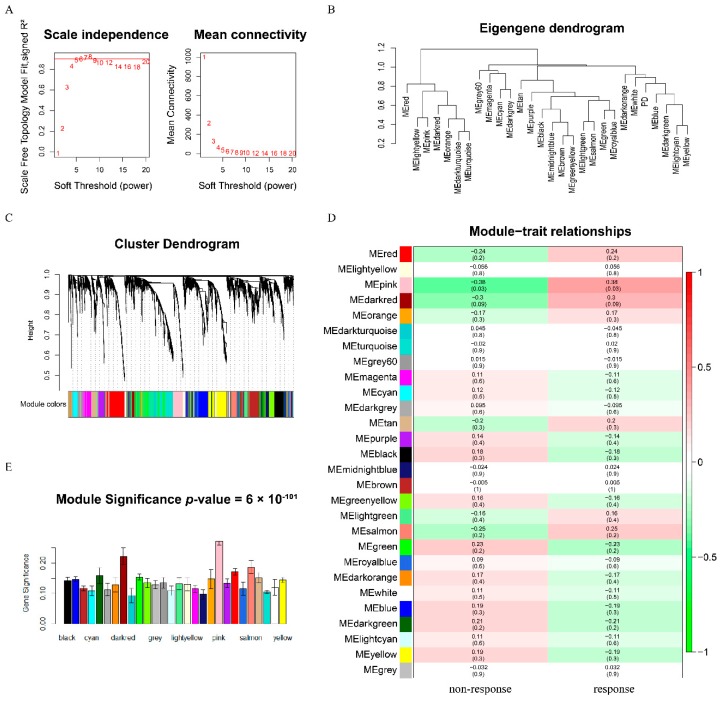
Identification of key modules connected with clinical features through WGCNA. (**A**) The left and right panel showed the scale-free fit index and the mean connectivity for various soft-thresholding powers, respectively. When the soft-thresholding powers (β) equaled five, the average degree of connectivity was close to zero. (**B**) The cluster dendrogram of module eigengenes. (**C**) The cluster dendrogram of 5,000 module eigengenes from the GSE91061 dataset. Each branch in the figure represented one gene, and every color below represented one co-expression module. (**D**) Heatmap of the correlation between module eigengenes and clinical traits of anti-PD-1 immunotherapy responsiveness. The color of cells in the heatmap represented the correlation coefficients of different sizes. Specifically, red colors represented the positive correlations and green colors stood for the negative correlations. The figure without brackets in each cell indicated the clinical feature correlation coefficients. The corresponding *p*-value was shown below in parentheses. The pink module was significantly correlated with response to anti-PD-1 therapy. (**E**) Distribution of average gene significance and errors in the modules associated with the response to anti-PD-1 therapy.

**Figure 3 genes-11-00435-f003:**
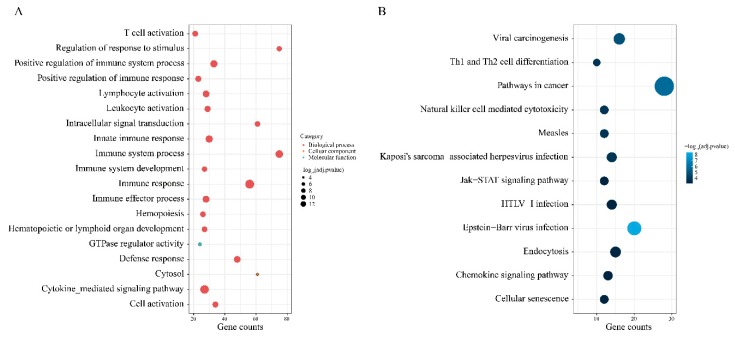
Enrichment analysis of gene ontology and Kyoto Encyclopedia of Genes and Genomes (KEGG) pathway for genes in the pink module. (**A**) Enrichment analysis of gene ontology. Annotation terms in the biological process, cellular component and molecular function were marked in red, yellow, and green, respectively. Bubble size represented the value of –log_10_-adjusted *p*-value of enrichment significance. (**B**) Enrichment analysis of KEGG pathway. Different colors represent the value of –log_10_-adjusted *p*-value of enrichment significance.

**Figure 4 genes-11-00435-f004:**
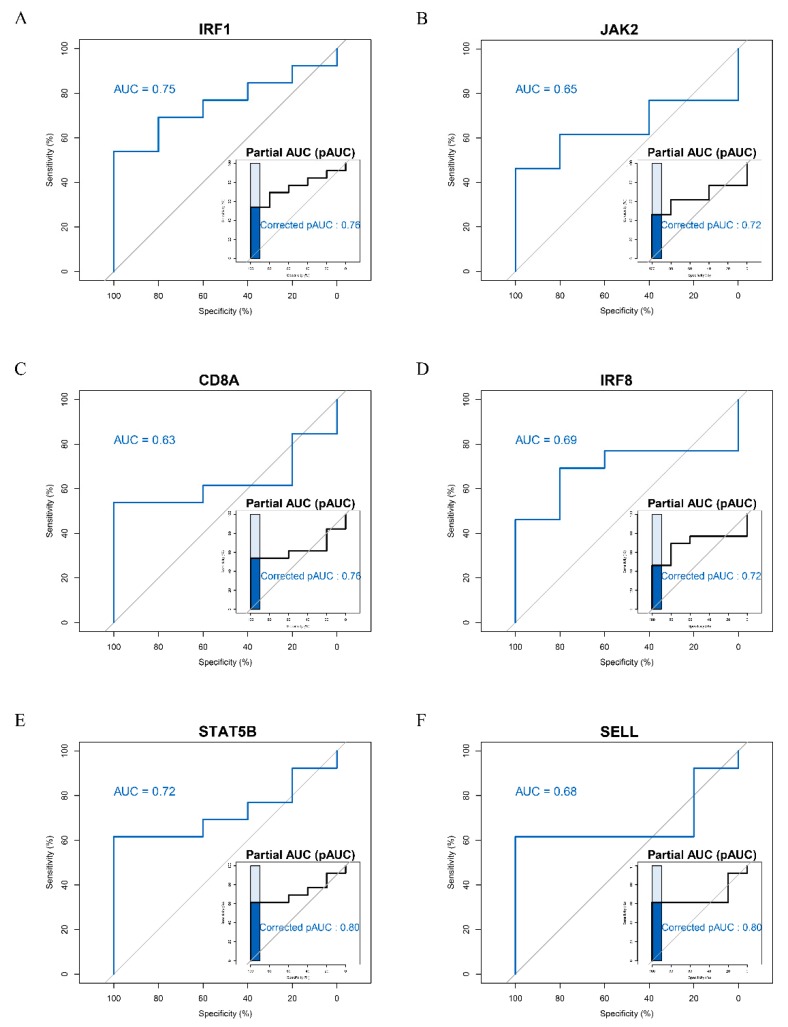
Receiver operating characteristic (ROC) curve analysis of six hub genes based on GSE78220. (**A**) *IRF1*, (**B**) *JAK2*, (**C**) *CD8A*, (**D**) *IRF8*, (**E**) *STAT5B*, and (**F**) *SELL.* The area under the ROC curve (AUC) and partial area under the curve (pAUC) are shown in each subgraph, and the pAUC is on the bottom right of the subgraph. The AUCs of *IRF1*, *JAK2*, *CD8A*, *IRF8*, *STAT5B,* and *SELL* were 0.75, 0.65, 0.63, 0.69, 0.72, and 0.68, respectively. The pAUCs of *IRF1*, *JAK2*, *CD8A*, *IRF8*, *STAT5B,* and *SELL* were 0.76, 0.72, 0.76, 0.72, 0.80, and 0.80, respectively.

**Figure 5 genes-11-00435-f005:**
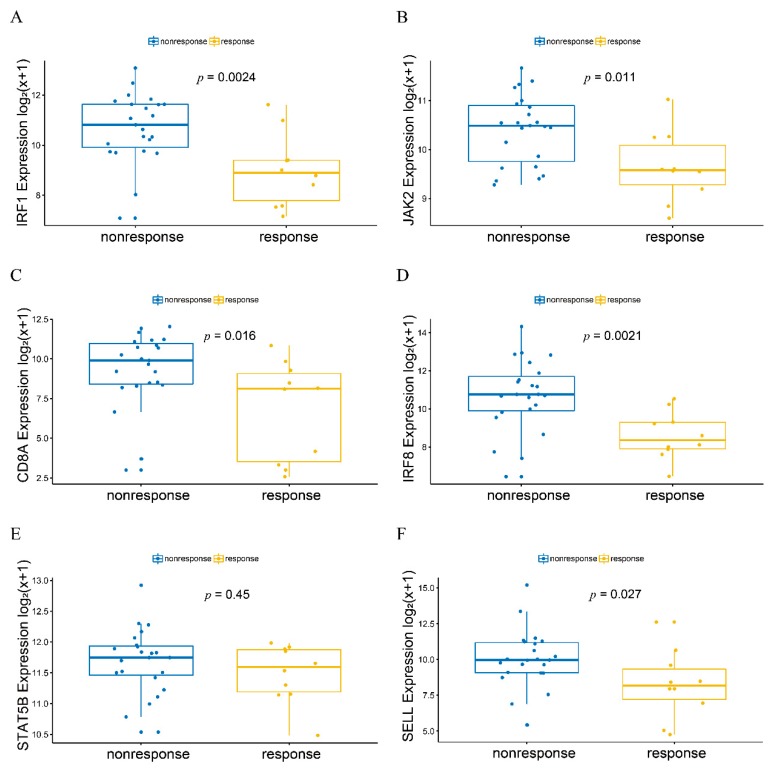
The expression of the six hub genes in responders or non-responders to anti-PD-1 therapy. (**A**) *IRF1*, (**B**) *JAK2*, (**C**) *CD8A*, (**D**) *IRF8*, (E) *STAT5B,* and (**F**) *SELL* gene expression differences between melanoma and normal tissues. The blue column represented the samples of non-responders, and the yellow column represented the samples of responders.

**Figure 6 genes-11-00435-f006:**
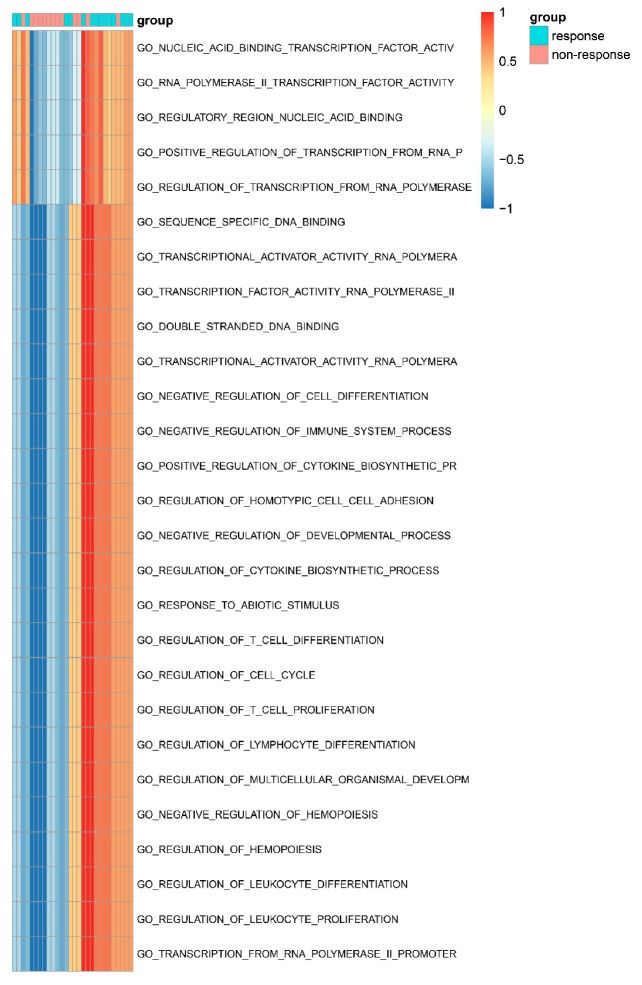
Gene set variation analysis of hub genes in the GSE78220 dataset. The functional analysis clustered gene ontology terms with *p*-value < 0.05 for six hub genes. The x-axis represents melanoma samples treated with anti-PD-1 therapy, including responders (green) and non-responders (red). The y-axis shows the enriched GO terms, and the orange ones indicate the up-regulation of the function, while the blue ones indicate the downregulation.

**Figure 7 genes-11-00435-f007:**
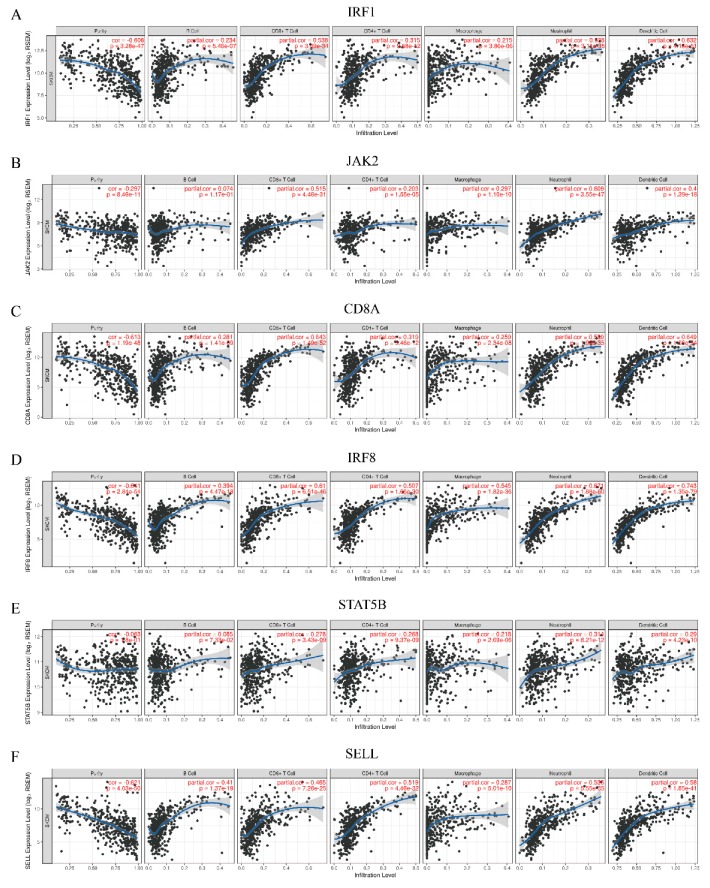
Correlation of six hub genes with immune infiltration in melanoma. (**A**) *IRF1*, (**B**) *JAK2*, (**C**) *CD8A*, (**D**) *IRF8*, (**E**) *STAT5B*, and (**F**) *SELL*. The x-axis demonstrates the immune infiltration levels. Spearman’s correlation coefficient and *p*-value are shown in the upper right corner.

**Figure 8 genes-11-00435-f008:**
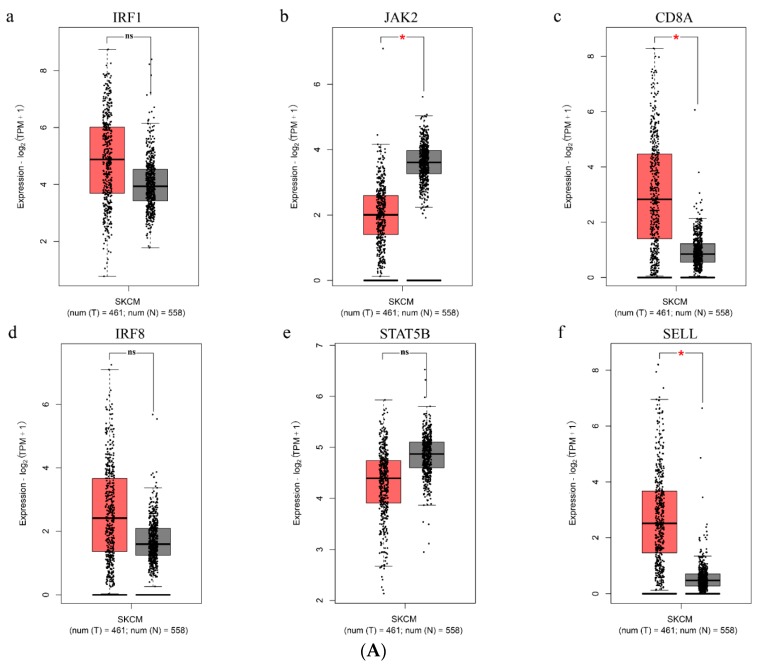
Validation of the expression level and prognostics in melanoma for six hub genes. (**A**) *IRF1*, *JAK2*, *CD8A*, *IRF8*, *STAT5B,* and *SELL* gene expression differences between melanoma and normal tissues. The red column represents the melanoma samples, and the black column represents the normal samples. (**B**) Survival analysis of *IRF1*, *JAK2*, *CD8A*, *IRF8*, *STAT5B,* and *SELL* in melanoma. The red line designates the samples with highly expressed genes, and the blue line indicates the samples with lowly expressed genes. (**C**) The multivariate Cox regression analysis of the six screened hub genes in overall survival. The horizontal axis (x-axis) represents time in days, and the vertical axis (y-axis) shows the probability of survival or the proportion of people surviving. The lines represented the survival curves of the two groups. ns, *p*-value > 0.05; *, *p*-value < 0.05.

**Figure 9 genes-11-00435-f009:**
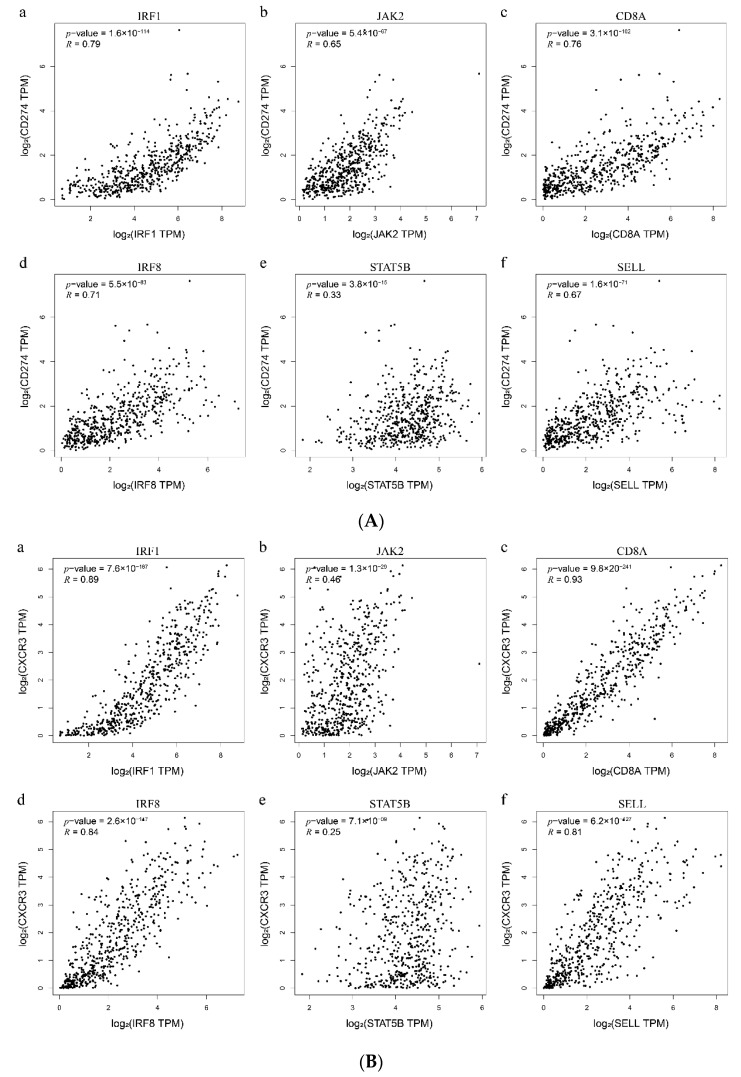
Expression association analyses between six hub genes and reported biomarkers for anti-PD-1 therapy in melanoma. (**A**) The relevance of PD-L1 (*CD274*) and six hub genes. (**B**) The relevance of *CXCR3* and six hub genes. (**C**) The relevance of IFN-γ (*IFNG*) and six hub genes. Six hub genes including *IRF1*, *JAK2*, *CD8A*, *IRF8*, *STAT5B,* and *SELL*. The x-axis and y-axis represent the expression level (log_2_TPM) of hub genes and reported melanoma biomarkers of anti-PD-1 therapy, respectively. The upper left corner of the picture shows the *p*-value and correlation coefficient calculated by the Spearman method. TPM: Transcripts per million.

**Figure 10 genes-11-00435-f010:**
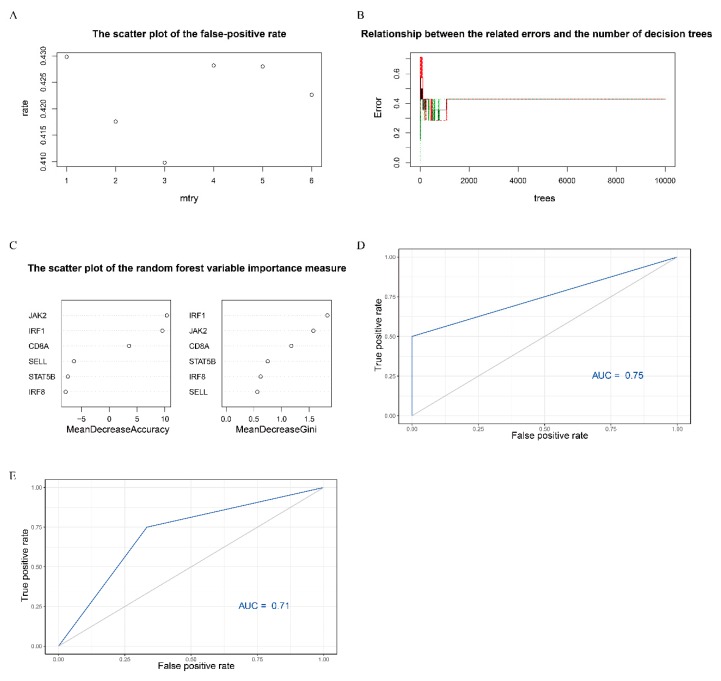
The correlation results of the random forest model. (**A**) The scatter plot of the false-positive rate. The vertical axis represents the false positive rate, and the horizontal axis represents mtry index. (**B**) Relationship between the related errors and the number of decision trees in random forests. The vertical axis represents the related errors, and the horizontal axis represents the number of decision trees. (**C**) The scatter plot of the random forest variable importance measure. The left and right panel were calculated based on the index of mean decrease accuracy and mean decrease Gini, respectively. (**D**) Evaluation of the prediction efficiency of the random forest model in immunotherapy response (GSE78220). (**E**) The validation set (GSE93157) verified the accuracy of the random forest model in immunotherapy response.

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
