# Peer review of "Identification of Potential Biomarkers for Anti-PD-1 Therapy in Melanoma by Weighted Correlation Network Analysis"

_genes, 2020, doi:10.3390/genes11040435_

Round 1

Reviewer 1 Report

The authors of Wang et al have analysed two datasets of melanoma samples following anti-PD-1 therapy, together with another independent dataset, to identify six hub genes that may be biomarkers for response to therapy. The findings of this study have the potential to be of interest to readers and to impact research in the area of prediction of response to anti-PD-1 therapy, but further information should be provided to ensure that these results are useful to other researchers.

MAJOR COMMENTS

  1. It is not clear to me how the authors chose their cut-offs regarding GO and KEGG pathway enrichment analysis (Section 2.4). Why 23 genes? Why the chosen p-value and gene count for the KEGG analysis? Are these arbitrary measures?

  2. The authors state that they used the Molecular Signature Database as a reference gene set. How do the authors ensure that they are not simply detecting genes that are generally dysregulated in melanoma rather than genes that are specifically dysregulated in response to anti-PD1 therapy? 

  3. It is not clear why the authors choose to only focus on the pink module for their study. The dark red and salmon modules seem to be similarly promising candidates for analysis. By only analysing the pink module, this approach appears to be at risk of ignoring useful information contained in other modules. Can the authors provide a similar analysis for other top candidate modules and include these information in the main text or supplementary information? Alternatively, can the authors justify their choice of only the pink module for analysis?
  4. How do the authors move from top 50 genes analysed in Cytoscape down to 13 genes (Section 3.3)? What were the criteria for inclusion/exclusion?

  5. The authors quote a "remarkable predictive ability" of pAUC > 0.7 (Section 3.4). How does this value compare to other biomarkers that have been identified in the literature cited in the Introduction? How do the pAUC values compare to traditional AUC values? Please provide this information in a main or supplementary figure.

  6. The authors state that the 6 genes are able to predict response to anti-PD-1 therapy. It is not clear how these genes actually predict response. Do the authors propose a linear model with weights for each gene expression value? How could a future researcher apply their predictions to their own dataset? Can the authors show boxplots indicating how these genes are up- or down-regulated in complete versus non-responders to anti-PD-1 therapy? I am entirely missing the method by which these genes can predict response.

  7. How were the patients in the TCGA cohort treated? This appears to be a major confounder with regard to the survival analysis. How do the authors avoid this limitation? 

  8. Can the authors comment more in the Discussion on the possible biological role of the six genes identified?

  9. Many of the figures contain text that is too small to read easily. For example, Fig 2a, Fig 2e and Fig 3. Please increase text size.

  10. The manuscript requires editing for language and grammar, as certain errors are present in the text that confuse its meaning.

MINOR COMMENTS

  1. The second line of the abstract contains a grammatical error.

  2. The authors should say what currency is being referred to when stating "$60,000" (pg 1).

  3. The first sentence of the second paragraph on pg 2 appears to be unfinished.

  4. Errors in the text appear in the 5th line of Section 2.1 ("standers"?), 3rd line of Section 2.6 ("particle" -> "partial") and second line of Section 3.8 (additional bracket). 

  5. Figure 2E and 2D are referred to out of order in the text.

  6. Figure 7a does not show the p-value (represented by an asterisk) for some panels in the figure. If not significant, this should be specified as "n.s".

  7. How did the authors define high and low expression of genes in Figure 7b?

Author Response

Response to Reviewer 1 Comments

      We appreciate the reviewers for their insightful and constructive comments, which immensely helped us to improve the quality of our manuscript. Our responses to the reviewers’ comments are described in the attachment, point-by-point.

      Please see the responses in the attachment. Thank you again for your nice suggestions.

Reviewer 2 Report

In the manuscript, Wang and Chai et al explored the clinically-relevant biomarkers in melanoma patients using weighted correlation network analysis, receiver operating characteristic, and gene set variation analysis. The obtained results were integrated with clinical outcomes. The study identified IRF1, JAK2, CD8A, IRF8, STAT5B and SELL and reported their important functional roles through in silico analysis. The authors highlighted that these six genes were implicated in immune system regulation and were associated with patients in response to anti-PD-1 immunotherapy. Although these six genes have the potential for biomarker development in anti-PD-1 therapy, the manuscript lacks clarify and the presented data requires more work.

Specific comments:

  1. In the Introduction, the cost is quite subjective. Is it in USD dollar? Suggest to leave it out.
  1. In the Abstract, the authors stated “Anti-PD-1 immunotherapy has shown dramatic clinical benefits, but it is failed to appear a response in some melanoma patients”. Could the authors please rephrase? My suggestion: “Anti-PD-1 immunotherapy has shown clinical benefits in improving patients’ overall survival, but some melanoma patients failed to response.”
  1. In Materials and Methods, the authors stated “…expression data of responders (complete response or particle response) or non-responders (progress disease)”. According to the RECIST guideline, it should be partial response (PR) and progressive disease (PD). Could the authors please clarify?
  1. In addition, patients with stable disease should be included in the study for biomarker discovery. Otherwise, the biomarkers that were identified in this study would be biased to a group of patients. Could the authors include patients with stable disease for their analysis. If the authors choose to exclude patients with stable disease, could the authors explain and state in the materials and methods section why they have exclude these patients?
  1. For correlation methods, Pearson and Spearman methods are used in the present study. Could the authors explain and state in the manuscript the reason(s) for choosing Pearson or Spearman during correlation analysis?
  1. In Result Section 3.1 and Figure 2, the coloured modules are not scientifically appropriate. Commonly, gene signature 1, 2, 3 etc. are labelled instead of labelling the types of colours. Could the authors clarify?
  1. In the Result Section 3.4, the authors claimed that the hub genes or functional signatures (IRF1, JAK2, CD8A, IRF8, STAT5B and SELL) were important to distinguish between responders and non-responders in anti-PD-1 therapy in melanoma. However, the ROC analysis was performed individually for the respective genes. If the authors claimed that the hub genes or gene signatures were important, these genes should be analysed as a group. Could the authors clarify?
  1. Functional enrichment analysis was also performed in Hugo et al. study. What are the findings in Wang and Chai et al. differ from the published findings in Hugo et al. paper? Additionally, the genes that were identified have already published in other studies. Could the authors please clarify the novelty?
  1. In Figure 7B, could the authors analyse the genes as a set and assess if the gene signatures improve the overall survival of the patients?
  1. Could the authors check that they have followed the reference list and citation guide for website and URL in the Genes Journal?
  1. In Figure 1, could the authors expand and elaborate figure caption?
  1. Could the authors please correct all the grammars and typographical errors in the article?

Author Response

Response to Reviewer 2 Comments

      We appreciate the reviewers for their insightful and constructive comments, which immensely helped us to improve the quality of our manuscript. Our responses to the reviewers’ comments are described in the attachment, point-by-point.

      Please see the responses in the attachment. Thank you again for your nice suggestions.

Round 2

Reviewer 1 Report

The authors have addressed many of the comments raised in the first stage of review. One remaining concern relates to the random forest model that was applied to the dataset. In order to correctly test the accuracy of the random forest model, it should be trained on one dataset (eg GSE78220) and then tested on an independent validation cohort. It is my understanding that the six hub genes were derived from GSE78220, and so the model should not be tested on this dataset. If the model accuracy is not high in an independent validation cohort, then this raises concerns as to whether the six hub genes are truly relevant to other studies. I recommend that the authors test their model on an independent cohort and report the model accuracy. 

Author Response

  Thanks again for reviewers’ valuable and insightful suggestions to help in improving our manuscript. Our responses to the reviewers’ comments are described below, point-by-point.

Point 1

In order to correctly test the accuracy of the random forest model, it should be trained on one dataset (eg GSE78220) and then tested on an independent validation cohort. It is my understanding that the six hub genes were derived from GSE78220, and so the model should not be tested on this dataset. If the model accuracy is not high in an independent validation cohort, then this raises concerns as to whether the six hub genes are truly relevant to other studies. I recommend that the authors test their model on an independent cohort and report the model accuracy.

Response 1

   Many thanks for your advice. According to the reviewer's suggestions, we have used the melanoma samples treated with anti-PD-1 therapy of independent dataset GSE93157 as the validation set to assess the accuracy of the random forest model. The code corresponding to the random forest model has been corrected in Supplementary Material S2. And the main text was modified in red as follows:

Method

2.11 Random Forest

A prediction model of anti-PD-1 immunotherapy response was constructed via the random forest classifier. The hub genes were the covariates of the prediction model. Random forest is a popular tool for classification and regression, which shows a powerful ability to construct a predictive model for new biomarkers. Random forest is less prone to over-fitting problems and can handle a large amount of noise. A random forest-based classifier was built via the randomForest package in R software based on the algorithm of Breiman and Cutler (Mach Learn,45(1):5–32, Oct 2001). The samples of GSE78220 (n = 28) (Cell, 165 (1), 35-44 2016 Mar 24) were randomly divided into the training set and test set via caret package, each of which contained 14 samples. Then, the decision tree model of the training set was established to obtain the classification. Next, average the classification results of each time to calculate the final classification. The model built by the training set would be tested by the test set. Each result would calculate the error rate through Out-of-bag (OOB) to evaluate the correct rate of the combined classification. OOB was the data not sampled when the training set was randomly sampled. The OOB samples were used to estimate the prediction error and variable importance (Pattern Recognition Letters,31;14,2225-2236 Oct 2010). Finally, the melanoma samples treated with anti-PD-1 therapy of GSE93157 (seven complete response or particle response samples and 11 non-response samples) (Cancer Res, 77 (13), 3540-3550 2017 Jul 1) were used as the validation set to verify the accuracy of the random forest model. AUC index was utilized to evaluate the efficiency of the prediction model.

Result

3.10 The random forest model of hub genes

We constructed a random forest classification model of anti-PD-1 immunotherapy response based on the screened six hub genes (Supplementary Material S2). During the process of building the random forest model, when mtry = 3, the false positive rate of model was the lowest (Figure 10A). The optimal model can be achieved when the number of decision trees was about 2000 (Figure 10B). Besides, the randomForest package provides two indexes to calculate the importance of variables. The one is the index to calculate the prediction error rate based on OOB and named mean decrease accuracy (%IncMSE). The other is to calculate the Gini coefficient based on the sample fitting model and named Mean Decrease Gini (IncNodePurity). The results showed that IRF1 and JAK2 were the more important variables in the prediction model (Figure 10C). Then, the AUC index was used to evaluate the efficiency of the model. The results showed that the prediction model had a good predictive ability for anti-PD-1 immunotherapy response (AUC = 0.75) (Figure 10D). Compared with the single gene, the random forest model had a better value of AUC except IRF1. Additionally, the samples of independent dataset GSE93157 (n = 18) were used as the validation set to verify the accuracy of the random model. The results also indicated that the random forest model can significantly distinguish the response to anti-PD-1 therapy for melanoma patients (AUC = 0.71) (Figure 10E).

Note: Figure 10E can be viewed in the attachment.

Reviewer 2 Report

NA

Author Response

Thank you for the help and support of the reviewer to help in improving our manuscript.